# Scalable Feature Compression for Edge-Assisted Object Detection Over Time-Varying Networks

**Zhongzheng Yuan** [1] **Siddharth Garg** [1] **Elza Erkip** [1] **Yao Wang** [1]

## ABSTRACT

Split-computing has recently emerged as a paradigm for offloading computation of visual analytics models from low-powered mobile devices to edge or cloud servers, by which the mobiles execute part of the model and compress and send the intermediate features, and the servers complete the remaining model computation. Prior feature compression approaches train different compression models and possibly visual analytics models to reach different target bit rates. We propose a scalable compression model that compresses the intermediate features of the YOLO object detection model into a layered bitstream, which can be easily adapted to meet the rate constraint of a dynamic network. Our approach achieves comparable rate-accuracy performance compared to prior non-scalable compression approaches over a large bitrate range.

## 1 INTRODUCTION

Large scale deep learning models have led to remarkable performances in various visual analytics tasks. The ability to run computationally intensive neural networks at high frame rate is pivotal for wide spread adoption of these models in applications such as autonomous driving, navigation for blind and visually impaired people, and augmented reality. However, the end-user mobile devices that captures the raw visual data are often limited in their computation power and unable to execute these models at real-time. Moreover, running these models consumes significant energy, which shortens the battery life of the mobile devices.

Recent works have proposed to address this problem through offloading the model computation to an edge or cloud server. One popular approach is to split the task network between the mobile and the server (Matsubara et al., 2022). The mobile device runs its part of the network and sends the compressed intermediate feature to the server. The server finishes the computation of the rest of the task network and sends the result back to the mobile device.

Using learned compression methods for compressing the intermediate feature (typically jointly trained with the task network) has shown promising results in achieving good trade-offs between the task performance and the bitrate. However, prior approaches train different compression mod-

els and possibly visual analytics models to reach different target bitrates. This requires the mobile device to store many different versions of the model in order to operate under a dynamically varying wireless network, which not only consumes large memory space but also requires switching between models, increasing the latency. Few works have explored using a single encoder model (including the mobile part of the task network plus the feature compression module) to achieve variable bitrates (Datta et al., 2022). To the best of our knowledge, no prior work has considered how to compress the intermediate features into a layered bitstream. With scalable or layered compression, the model can efficiently adapt to changing bandwidth, by generating different number of layers. The layered bit stream can be further paired with unequal error protection, so that the lower layers (more important) can be better protected with guaranteed delivery and consequently a base level of task performance.

In this work, we propose a scalable compression model for split computing. We generate a layered bitstream by transforming the original intermediate feature tensor into multiple groups of feature channels; The first group forms the base layer operating at the lowest rate with a corresponding low analytics performance, and each subsequent group forms a higher enhancement layer. Adding each enhancement layer to the previous layers leads to an increased bitrate and improved analytics performance. We implement this scalable compression framework for the task of object detection using the YOLO model (Redmon et al., 2016) as the task network. We further investigate two options. In Option 1, both the mobile and server parts of the task network are fixed (jointly trained with the compression part)

---

[1]Department of Electrical and Computer Engineering, New York University, New York, USA. Correspondence to: Zhongzheng Yuan <zy740@nyu.edu>.

*Proceedings of the 6th MLSys Conference Workshop on Resource-Constrained Learning in Wireless Networks*, Miami, FL, USA, 2023. Copyright 2023 by the author(s).

regardless the target bit rate (or how many layers are sent). In Option 2, the server part of the task network is allowed to change, depending on the target bit rate range. Option 2 achieved comparable rate-detection performance with the non-scalable approaches over a large bit rate range, while having the benefits of scalability.

## 2 RELATED WORKS

### 2.1 Split Computing with Feature Compression

A main challenge in achieving good task performance in split-computing model is designing the method for compressing the intermediate feature at the point of split. Several works have proposed using learned compression methods for compressing the intermediate feature at the split point. Learned feature compression model with hyperprior network was proposed by (Singh et al., 2020). The intermediate feature at the point of split was quantized and entropy coded using a hyperprior network. The model had to be trained end-to-end so that different networks have to be trained for each bitrate. Moreover, the model considered a very late split point, so that most computation still has to be performed by the mobile.

A split-computing model was proposed for object detection in (Yuan et al., 2022), which considered early split-point to reduce the mobile run-time. Light-weight spatial and channel reduction was proposed to reduce the feature dimension and hyperpriors are used to facilitate the entropy coding of reduced features. The proposed split computing model achieved significant speed up, while achieving better rate-accuracy performance than benchmark methods such as compressing the image using learnt compression at the mobile, and decompressing the image and running the object detection at the server. But because the models were trained end-to-end separately for each rate point, the mobile device and server both need to store and operate a collection of models in order to operate under a variable rate channel.

In (Datta et al., 2022), a feature dimensionality reduction layer was also proposed, and entropy coding was performed through Huffman Coding. The paper proposes to swap only the compression modules for variable rate compression, while keeping the task networks fixed. Nevertheless, the paper's results show that training only the compression modules sacrifices the accuracy-rate performance quite substantially compared to end-to-end training.

### 2.2 Scalable Compression

Scalable image compression (also known as layered compression or coding) encodes an image into a base layer $z_1$ and additional enhancement layers $z_2$, $z_3$, ..., $z_M$ for a total of $M$ layers. The layers are embedded in that layer $m$ is useful and contributes to improved quality only if all previous layers up to $m - 1$ are available.

In the case of feature compression for analytics, scalability refers to generate a bitstream with multiple layers, so that each additional layer leads to improvement in the analytics performance. In practical applications, the mobile device is often deployed in areas with weak and unstable internet connection. In such scenarios, scalable compression can be particularly useful. In the case of a sudden drop in the network throughput (e.g., switching from 5G to 4G wireless network), the base layer can be transmitted to ensure a basic performance in analytics. When the network condition improves, additional layers can be generated and sent.

There have been several works on learned scalable compression of images for human visualization. In (Jia et al., 2019), scalability is achieved by using multiple encoder networks to successively compress the residual of the reconstructed image and send the residual latent information in layers. Another model propoesd by(Mei et al., 2021) encodes the input image to layered latent features and uses lower layer latents to predict and enhance the higher layer bitstreams. More recently, a fine grained scalable model is proposed by (Ma et al., 2022). The model generates a base and an enhancement feature tensor. The base feature is sent as a whole, while the enhancement feature tensor is split along the channel dimension, and each channel is sent one-by-one for each enhancement layer.

Several prior works have considered scalable compression for analytics, such as (Yan et al., 2020), and (Choi & Bajić, 2022). But the scalability proposed by these works refers to the ability of the server network to perform additional analytics tasks as it receives each additional bitstream, while our proposed scalable model aims at increasing the accuracy of a single task with each additional layer. To the best of our knowledge, this work is the first to propose a scalable feature compression model for analytics task.

## 3 PROPOSED METHODS

### 3.1 Split-Computation Model

The task network is split at an intermediate point of the network into two parts, the first part is ran on the mobile device and the second part on the server. We will refer to the mobile part of the network as the task encoder $\mathcal{F}(x)$ and the server part of the network as the task decoder $\mathcal{G}(x)$. The intermediate feature at the output of the task encoder $y = \mathcal{F}(x)$ is compressed and delivered to the server, which is then decompressed and sent to the task decoder for inference.

Choosing the point of split is an important consideration, because it affects the amount of computation needed to be done by the mobile device. In order to keep the computation at the mobile side low, we choose a split point that is

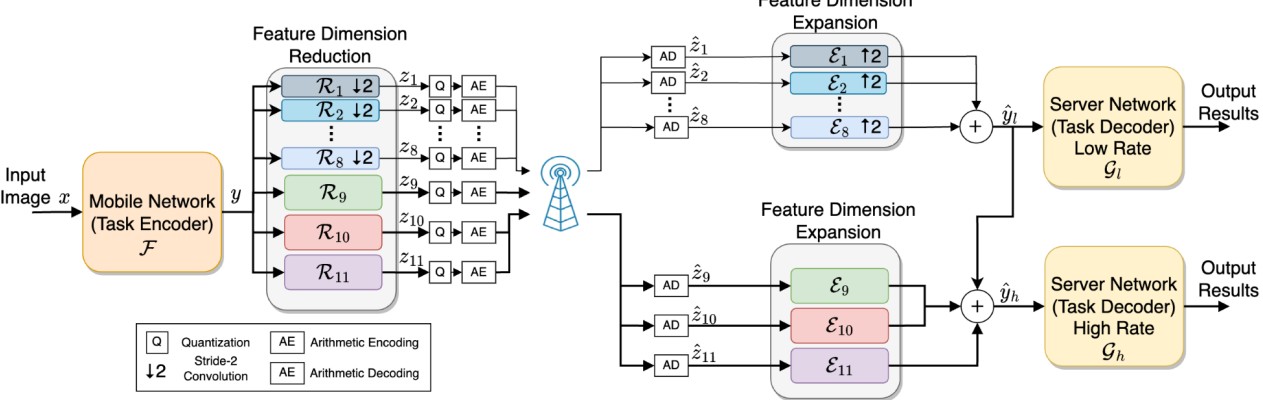

*Figure 1.* Overview of the proposed model. The task network is split into two parts that separately runs on the mobile device and the server. The model generate different layers by generating and entropy coding different groups of reduced feature channels from the task encoder. In this example, the original feature $y$ with 256 channels is first reduced into 4 tensors of 8 channels each. The first tensor is further split into 8 tensors of 1 channels each for the low-bitrate region, forming a total of 11 layers. The model uses different task decoders depending on the number of layers received.

relatively early in the network. For the YOLOv5 network, we experimented with the point of split D4, which is the point after the fourth downsample convolution.

## 3.2 Feature Dimension Reduction and Compression for Non-scalable Compression

The feature $y$ typically consists of a large number of channels. While this is beneficial for the task performance, there is significant redundancy among the channels. Coding these channels directly and independently are not efficient. Following the approach in (Yuan et al., 2022), we use a 1x1 convolution layer to reduce the channel dimension of the feature tensor before performing entropy coding. Furthermore, we found that for the lower scalable layers, it is beneficial to also reduce the spatial redundancy of the features by downsampling the features in the spatial dimensions. Therefore, for the lower scalable layers, we use a 5x5 stride-2 convolution layer to reduce both the channel and spatial dimension of the feature tensor. Specifically, the reduction layer takes the feature $y$ with $N$ channels and generates the reduced feature $z = \mathcal{R}(y)$ with $N_R$ channels. For the lower scalable layers, $z$ will be downsampled to half the spatial resolution of $y$.

The reduced feature tensor is quantized through $\hat{z} = \lfloor z \rceil$ and encoded into a bitstream using an arithmetic encoder. Following the hyperprior image compression model proposed by (Ballé et al., 2018), we train a hyperprior network which generates a side bitstream for estimating the mean and variance of the to-be-coded features.

## 3.3 Generating Scalable Feature Layers

To achieve scalability, we transform the original feature tensor with multiple feature channels into groups of channels, each with a small number of channels, to form different scalable layers. We draw inspiration from principal component analysis (PCA) to perform dimensionality reduction along the channel dimension through linear transform. Essentially each layer consists of a subset of transform coefficients. But whereas the transform in PCA is determined to minimizes the MSE of the reconstructed signal, our model is trained end-to-end to minimize the rate-detection loss.

We first perform dimensionality reduction to the intermediate feature at the split point to reduce the features into $M$ groups of features $z_i = \mathcal{R}_i(y), i = 1, 2, \ldots, M$, each with a small number of channels. Unlike the single dimensionality reduction layer in the non-scalable model, the scalable model uses $M$ separate convolution layers $\mathcal{R}_i, i = 1, 2, \ldots, M$, to generate $M$ groups of features $z_i = \mathcal{R}_i(y)$ with $N_{R_i}$ channels. A separate hyperprior network is trained to estimate the mean and variance parameters for each $z_i$, which are individually quantized and entropy encoded using their respective hyperpriors.

At the server side, $M$ separate dimension expansion layers $\mathcal{E}_i$ are used to expand all received dequantized features $\hat{z}_i$ back to $N$ channels and the original spatial dimension of $y$. The expanded tensors are added together to produce the input to the task decoder. During inference, depending on how many scalable layers are sent and received, the recovered feature $\hat{y}$ is the sum of all expanded tensors:

$$\hat{y} = \sum_{i=1}^{m} \mathcal{E}_i(\hat{z}_i), m \in \{1, 2, ...M\}, \quad (1)$$

where $m$ is the number of scalable bitstreams received. This combined feature is then input to the task decoder network to produce the analytics result $t = \mathcal{G}(\hat{y})$.

We noticed that using a single task decoder for a large bitrate

range will lead to suboptimal performance. Since computation and storage is less constrained on the server side, the server can run multiple decoder networks for different bit rate ranges. We therefore use one task decoder network $\mathcal{G}_h$ for the high rate range and another task decoder network $\mathcal{G}_l$ for the low rate range. The task encoder network and the feature dimension reduction layers is shared for both bitrate ranges. An overview of our scalable model is shown in Figure 1.

## 3.4 Training

The task model (including the task encoder $\mathcal{F}$ and decoder $\mathcal{G}$) is first initialized with pre-trained weights that were trained on the entire dataset without compression. Then we follow a three-step approach for converting the model for split-computing and training the model end-to-end for rate-task performance.

Instead of using the rate-distortion loss as for learnt image compression in (Ballé et al., 2017), we minimize the following rate-task loss, following (Yuan et al., 2022):

$$L = L_{Rate} + \lambda \cdot L_{Task}$$
$$L_{Rate} = \mathbb{E}_{x \sim p_x}[-\log_2 p(\hat{z})] \qquad (2)$$
$$L_{Task} = L_{coord} + L_{obj} + L_{class}$$

The rate loss $L_{Rate}$ measures the rate needed to encode the quantized tensor $\hat{z}$, the task loss $L_{Task}$ measures the performance of the task network, and the hyperparameter $\lambda$ controls the trade-off between rate and task performance. In the case of object detection, the task loss is defined as the combination of bounding box coordinate loss $L_{coord}$, objectiveness loss $L_{obj}$, and object class loss $L_{class}$ proposed by (Redmon et al., 2016). The above loss considers a single feature tensor $z$. In the following, we discuss how to train the multiple embedded feature layers $z_i, i = 1, 2, \ldots, M$.

### 3.4.1 Pre-Training of Feature Reduction and Expansion Modules

The pre-trained task network assumes that the original feature $y$ generated by the task encoder is received intact at task decoder. Ideally, the reconstructed features $\hat{y}$ from the reduced features $z_i$ should be similar to the original feature $y$. Therefore, we pre-train the feature reduction and expansion modules, $\mathcal{R}_i$ and $\mathcal{E}_i$, by minimizing the mean square error between $y$ and $\hat{y}$ in the absence of quantization of $z_i$.

### 3.4.2 Multi-Round Refinement of Scalable Layers Using Rate-Task Loss

We then refine the model end-to-end with the rate-task loss using a training strategy that updates all scalable layers iteratively. For each batch of training data, we input the batch into the network over $M$ rounds. In round $l$, only layers 1 to $l$ are activated, for $l$ from 1 to $M$. The network is

updated using the loss corresponding to having only layers up to $l$:

$$L_l = \sum_{i=1}^{l} (L_{Task}^i + \lambda_i \cdot L_{Rate}^i), \qquad (3)$$

where $L_{Task}^i$ and $L_{Rate}^i$ are the task loss and rate loss respectively when using scalable layers up to the $i$th layer. Note that the rate term for the $i$th layer includes the rate for the layers 1 to $i$. After going through $M$ rounds, the compression and decompression modules corresponding to all layers will be updated. We experimented with different $\lambda_i$ values for training and reported the best performing model. We found that using $\lambda_1 = 0.1, \lambda_i = \frac{\lambda_1}{i}$, for $i > 1$ led to good results.

Note that with this strategy, the modules for the lower layers are updated more times than the higher layers. This is appropriate because of the embedded nature of the layered bit stream. We have found that this training strategy yields better performance over the entire rate range than some alternative approaches, including progressive training, where we first train only the base layer modules, and then train the second layer, while fixing the base layer, and so on.

### 3.4.3 Separate Decoder Training

Initially we used the strategy in Sec. 3.4.2 to directly train $M$ layers with each layer having the same number of channels. But we found that the rate-detection performance is always lacking at some part of the entire rate range when compared to the performance of non-scalable coding. For example, with 4 layers each with 8 channels, the performance is good at the higher rate range including at least two layers, but the performance is poor at the lower rate range including only the first layer. In order to solve this problem, we tried to split the first layer (8 channels) into 8 layers (each with 1 channels), yielding a total of 11 layers. This still did not lead to satisfactory results at the lower rate range. We suspect that it is hard for the same task encoder and task decoder to perform object detection accurately over the entire rate range. Therefore, we choose to train a different task decoder when less than four layers (each 1 channels) are delivered, while the task encoder is fixed (see Fig. 1).

More generally, starting with a trained model that generates $M$ layers, the original base feature tensor that consists of $N_{R_1}$ channels is subdivided into $M'$ tensors with $N'_{R_1}$, $N'_{R_2}$, ..., $N'_{R_{M'}}$ channels respectively. This has the effect of splitting the original base layer into $M'$ scalable layers for the lower bitrate range, and the entire model will consist of $M + M' - 1$ scalable layers. We use the same training loss in Eq. 3 for training the expansion modules for layers 1 to $M' - 1$ and a single low rate task decoder, to be used when only up to $M' - 1$ layers are delivered. For this training stage, the task encoder, channel reduction and expansion

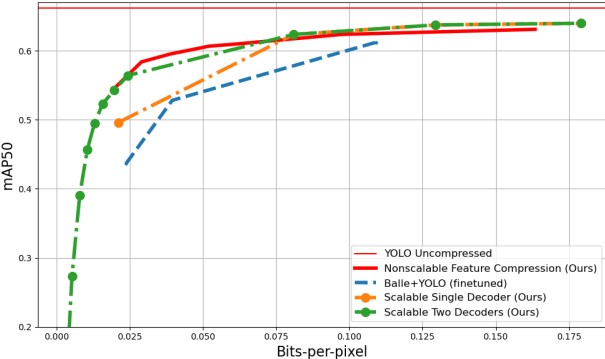

*Figure 2.* Rate vs. Detection Accuracy on the COCO-Traffic dataset. Only the scalable compression models allow for running a single task encoder for all the rate points.

layers, and the hyperprior networks for the high rate scalable layers are not updated. This makes the performance of the high bitrate scalable layers unchanged.

## 4 RESULTS

### 4.1 Experimental Setup

We used the Ultralytics YOLOv5 implementation (Ultralytics, 2020) as the basis of our split-computing model. The smaller-sized YOLOv5s model with 7.2 million parameters was chosen for faster training and inference speed. The model was initalized with weights provided by the implementation, which were trained with images from the entire COCO training set (Lin et al., 2014). We used the same strategy of data augmentation as in the pretrained model. All training and test images were resized to $640 \times 640$ as input to the model.

Instead of detecting all the classes in the COCO dataset, we consider a case where the detection network is focused on detecting 9 classes of objects that are relevant to traffic applications: Person, Car, Bus, Truck, Motorcycle, Traffic Light, Fire Hydrant, Stop Sign, and Parking Meter. The detection heads of our model is modified to detect only these classes, and we train the model using the subset of COCO that contains these classes.

### 4.2 Rate vs. Detection Accuracy Performance

We compare our results to two non-scalable baselines. The first one is the non-scalable split computing approach proposed in (Yuan et al., 2022). With the second baseline, the mobile compresses an image using the learned image compression model by (Ballé et al., 2018), then the server runs the object detection model on the decompressed image. With both baselines, different task modules and the compression/decompression modules are trained using the rate-task loss for different rate points.

The rate vs. detection accuracy performance is shown in Figure 2. The detection accuracy is measured in mAP50, or mean-Average-Precision using a 0.50 IOU bounding box threshold. We show results from two options of the proposed scalable compression approach. The first one uses a single task decoder regardless of the number of layers sent, trained directly with steps described in Section 3.4.2. This option used 4 layers, each with 8 channels. The second option uses the same encoder as option 1, but split the base layer of option 1 into 8 layers of 1 channel each, as shown in Fig. 1. And an additional decoder is trained using the steps described in Section 3.4.3. The two-decoder approach achieved higher performance over the lower bitrate range than the single-decoder model.

With the two-decoder scalable model, our results is comparable to the non-scalable split-computing method throughout the entire bitrate range, and even managed to outperform the non-scalable models at high bitrate range. We suspect that this is due to the non-scalable model have not searched for the optimal hyperparameter (lambda, number of downsamples, and channels) during training.

Note that for all the compared baseline methods, different points on the curve require a different set of task encoder, task decoder, compression and decompression modules. On the other hand, our scalable compression approach achieves all the rate points with a single task encoder. This makes our scalable approach much more practical for real-world applications. Additionally, the use of scalable bitstream means that the sender can adaptively generate and send layers at a rate that matches with the estimated network throughput. Furthermore in the case that some enhancement layer packets are dropped, the task decoder can still rely on the received base layer and lower enhancement layers to perform object detection, albeit at a lower accuracy.

## 5 CONCLUSION

In this paper, a scalable feature compression approach for split-computing is proposed. Our approach utilizes multiple channel reduction modules to generate different layers of the feature tensor to achieve rate scalability. The performance of our model is comparable to non-scalable approaches, while enjoying the benefits of scalable bitstream and requires a single task encoder. Although we only demonstrated the performance of the proposed approach on the COCO-Traffic dataset, we expect similar performance to hold for the full COCO dataset. The current approach generates different feature layers by simple linear transform with a single convolution layer. Our future research will explore more sophisticated methods to generate the different layer features and reduce the redundancy between different layers.

## ACKNOWLEDGEMENTS

This work was supported in part by NSF grant 2003182, and by Intel. In addition, Yuan and Wang were also supported in part by NSF grant 1952180.

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

## A    DETECTION RESULTS FOR DIFFERENT SCALABLE LAYERS

In Figure 3, we show some examples of images with bounding boxes that were detected using the scalable model. When only the base layer is available, the model is able to detect some objects in the scene. Although the model can confuse some object classes such as Truck (labeled as 3) and Car (labeled as 1), the results would still be useful for applications such as obstacle avoidance. As the number of layers increase, the number of correctly detected objects in the image increase, and some objects that were previously incorrectly classified are corrected.

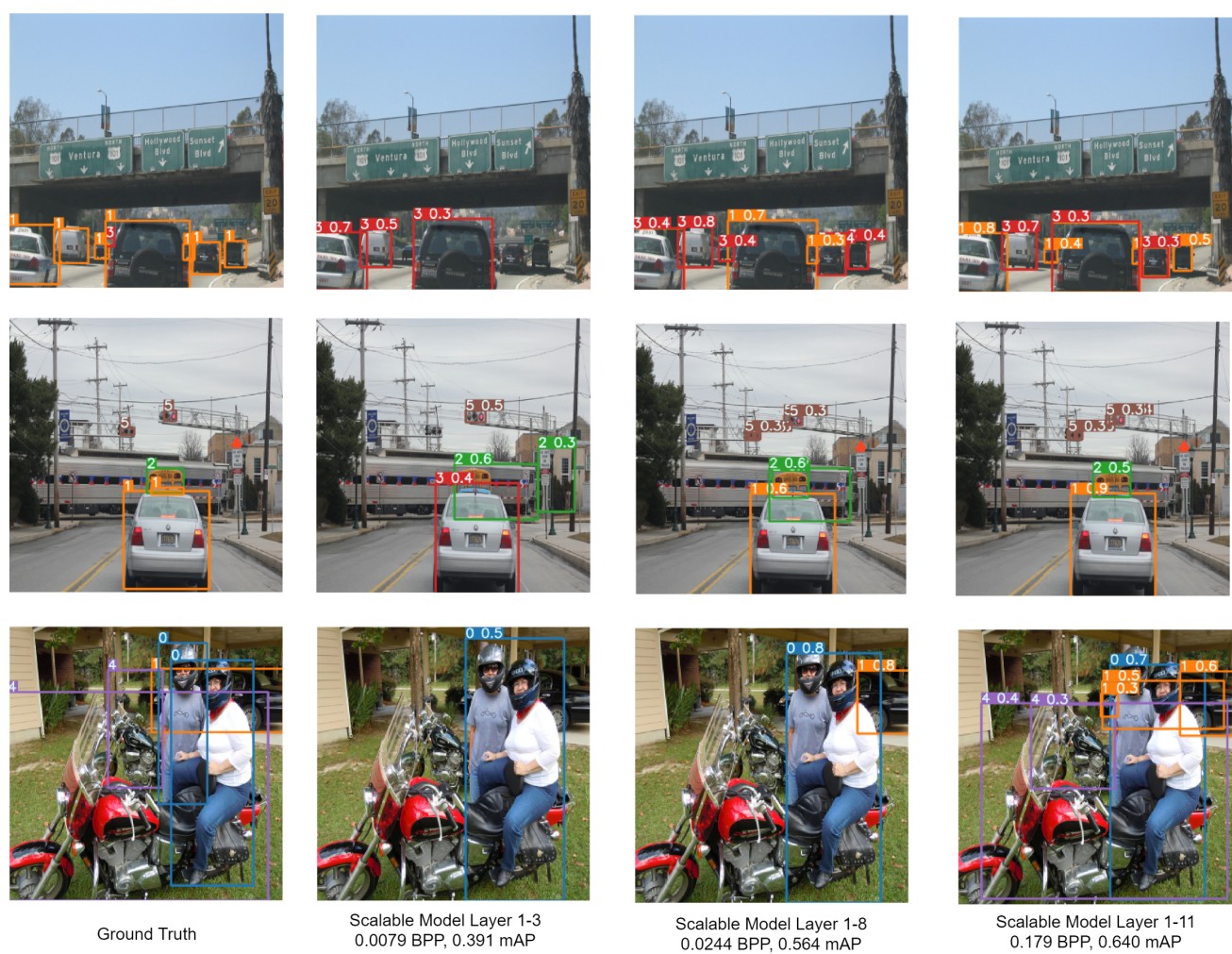

*Figure 3.* Examples of detection results for different scalable layers.