# OpenReview forum: "Scalable Feature Compression for Edge-Assisted Object Detection Over Time-Varying Networks"
_MLSys/2023/Workshop/RCLWN — MLSys-RCLWN 2023_

### Official Review · Reviewer_YJ6S · 2023-04-13
**The paper solves an important problem in remote classification but lacks some details in the proposed solution**

**Rating:** 6
**Confidence:** 5

**Review:**

+ The proposed scalable compression require that all lower rate layers be received for any upper layers to be useful. In practice, the variability of the channel means that it is highly likely for lower rate layers to be lost as well. How does the author propose to solve this issue?
+ More details needed on how PCA is performed for the task objective function.
+ Why is pretraining of the uncompressed network needed? Shouldn't the loss function take care of the trade off between rate and accuracy?
+ Why is MSE the distortion function used to pretrain the compression network? As the authors have emphasised in this paper, the goal should be the end objective function rather than MSE distortion.
+ How does the complexity compare between different baselines and your proposed scheme? What if the mobile device uses a handcrafted image compression scheme, such as BPG?

---

### Official Review · Reviewer_FeAL · 2023-04-29
**Well written paper on an interesting problem, but with limited technical novelty and limited numerical experiments.**

**Rating:** 6
**Confidence:** 5

**Review:**

The paper proposes a scalable feature compression framework for split learning. The idea is to compress the features into multiple code rates, resulting in increasing quality reconstruction of the feature vector, and hence, different accuracies at the receiver,. These levels can then be chosen based on the available link capacity. The scalable nature of the encoding scheme allows the edge device to employ a single encoder, rather than picking from a set of encoders depending on the link quality. Overall, the problem is well motivated, and the proposed solution is reasonable . The paper is well written and easy to follow.

On the other hand, the contribution, from a technical point of view, is quite limited. The problem is not much different from scalable image compression, which has been widely studied. This is already possible with current image compression codecs, and there has been several papers proposing neural-based scalable compression schemes. The paper can be considered as a direct application of these ideas for the compression of feature vectors in split learning.

In particular, the following papers study scalable image compression:

C. Jia, Z. Liu, Y.Wang, S. Ma, and W. Gao, “Layered image compression using scalable auto-encoder,” in Proc. IEEE Conf. Multimedia Inf. Process. Retrieval, 2019.

C. Cai, L. Chen, X. Zhang, and Z. Gao, “Efficient variable rate image compression with multi-scale decomposition network,” IEEE Trans. Circuits Systems Video Tech., vol. 29, no. 12, pp. 3687–3700, Dec. 2019.

Y. Mei, L. Li, Z. Li and F. Li, "Learning-Based Scalable Image Compression With Latent-Feature Reuse and Prediction," in IEEE Trans. on Multimedia, vol. 24, pp. 4143-4157, 2022.

Y. Ma, Y. Zhai and R. Wang, "DeepFGS: Fine-Grained Scalable Coding for Learned Image Compression, arXiv preprint  2201.01173, 2022.

This line of research should definitely be mentioned and reviewed in the paper. The above papers actually propose more advanced coding schemes, not limited to feature grouping, where each layer can be a compressed version of all the features, while the higher layers simply refine the quality of the reconstructed feature vector. I believe these may provide a better trade-off, and should be considered by the authors, at least for comparison.

Additionally, I find the presented numerical experiments rather limited. There is a single result for a single dataset/ single task for a single split point. Also, the focus has been on the higher compression rates. The authors claim ".. with only moderate accuracy drop in the low-bitrate regime", but this is not accurate, as there is a significant drop in the performance. Hence, one wonders, if more advanced techniques used in the above papers, including using more than two decoder networks, can fill that gap. It would also be good to have results for other split points, and see if different trade-offs can be observed.

Finally, one can also consider directly using scalable image compression codecs (see the above references for details) as another benchmark, to compare with the proposed learned approach (applying these to the feature vector, not to the image).

---

### Official Review · Reviewer_8YRy · 2023-04-29
**This paper proposes a scalable compression model that compresses the intermediate features into layered bitstreams. By using this approach, the memory cost on the mobile devices as well as the switching time are reduced. This is a new and interesting approach that is more practical for real-world applications. As these model deployments become more common, the problem is quite important.**

**Rating:** 7
**Confidence:** 3

**Review:**

Pros -

The paper's contribution is straightforward, and the problem is well presented.
It is impressive to see the empirical results.
Overall, the paper is well written, if somewhat verbose.

Cons -

There were several design decisions that went into the method, and they are not ablated, it would have been interesting to see these.
Due to the compute constraints of the mobile device, an efficiency analysis would have been helpful.

---

### Meta-Review · Area_Chair_44KG · 2023-05-04

**Recommendation:** Accept
**Confidence:** 4

**Metareview:**

The Reviewers point out both a number of positive, novel contributions of the paper, but also point out a few short comings. Mainly, they would like to see more experimental results (including ablation studies) to validate their assertions and findings.

Overall I think the paper presents a solid contribution, but could additional refinement (if edits are allowed) to address some of the reviewers concerns.